# Gliadin-Rich Diet Worsens Immune and Redox Impairments in Prematurely Aging Mice

**DOI:** 10.3390/cells14040279

**Published:** 2025-02-14

**Authors:** Estefanía Díaz-Del Cerro, Antonio Garrido, Julia Cruces, Noemí Ceprián, Mónica De la Fuente

**Affiliations:** 1Department of Genetics, Physiology, and Microbiology (Unity of Animal Physiology), Faculty of Biology, Complutense University of Madrid (UCM), 28040 Madrid, Spain; jcruces@pdi.ucm.es (J.C.); nceprian@ucm.es (N.C.); mondelaf@ucm.es (M.D.l.F.); 2Institute of Investigation 12 de Octubre (i+12), 28041 Madrid, Spain; 3Nanocaging Research Group, Department of Biosciences, Faculty of Biomedical and Health Sciences, Universidad Europea de Madrid, 28670 Madrid, Spain

**Keywords:** gluten, gliadin-rich diet, prematurely aging mice, immune function, oxidative state, peritoneal leukocytes, organs, celiac disease

## Abstract

Gliadin is one of the most important fractions of gluten, a glycoprotein closely linked to the development of negative effects on physiological functions and the development of gastrointestinal diseases, such as celiac disease (CD). Research suggests that inadequate stress responses and anxiety states may trigger or at least contribute to the development of these pathological conditions. Peritoneal leukocytes from Prematurely Aging Mice (PAM), which are chronologically adult mice with compromised responses to stress and anxiety, exhibit functional changes when exposed in vitro to gliadin peptides, resembling some immune alterations found also in CD patients. This observation prompted us to investigate the effects of a gliadin-rich diet on immune function and redox state in PAM. In this study, adult female PAM were fed either a gluten-enriched diet (PAMD, 120 g/kg) or a standard diet (PAMC) for four weeks. Immune function parameters in peritoneal, splenic, and thymic leukocytes (phagocytosis, chemotaxis, Natural Killer activity, lymphoproliferation) and redox markers (glutathione reductase, glutathione peroxidase, reduced/oxidized glutathione, xanthine oxidase activity, lipid peroxidation) were evaluated. The results showed that PAMD exhibited more impaired immune function, lower antioxidant enzyme activities, and reduced glutathione concentrations, as well as higher oxidized glutathione and increased xanthine oxidase activity compared to PAMC. These findings suggest that a gliadin-rich diet worsens immune and redox impairments in PAM, resembling some of the alterations previously described in CD, and indicating the potential of this animal for studying gluten-induced immune dysregulation.

## 1. Introduction

Gluten, a glycoprotein found in grains such as wheat, rye, and barley, consists of two main fractions: gliadin and glutelin, both characterized by a high content of proline (Pro) and glutamine (Gln) residues. Gliadin is the primary trigger of immune responses associated with certain pathological conditions [1,2,3,4]. In other grains, prolamins analogous to gliadin, such as secalin (rye), hordein (barley), and avenin (oats), also contain protein sequences harmful to sensitive individuals [5]. These prolamins resist proteolytic degradation in the intestinal tract, leading to their accumulation and persistence in the intestine. Consequently, they provoke a dysregulated immune response, resulting in mucosal remodeling with atrophy, hyperplasia, and lymphocytic infiltration, which contribute to disease symptomatology [6].

The consumption of gliadin has been linked to various negative effects. Exposure to gliadin can impair immune function by reducing NK cell activation, a crucial defense mechanism against malignancy, suggesting that gluten might act as a carcinogen in susceptible individuals [7]. Additionally, gliadin induces IL-8 production, which promotes neutrophil migration, mimicking the chemoattractant effect of *E. coli* and potentially amplifying inflammatory responses [8]. As a result, inflammation within the intestine may extend systemically, leading to cytokine imbalances that affect multiple physiological systems, including the nervous and endocrine systems. These disruptions trigger the release of additional neurotransmitters and hormones, leading to widespread physiological alterations [9,10]. At the cellular level, gliadin peptides can also penetrate cells and activate free radical production, leading to oxidative stress, lipid peroxidation, and cellular damage [11,12,13,14,15]. Indeed, oxidative stress correlates with increased lipid hydroperoxides, nitric oxide, and decreased antioxidant defenses such as glutathione (GSH) concentrations and glutathione reductase (GR) activity [16].

A diet rich in gluten can also contribute to the onset of various pathologies, including chronic inflammatory and oxidative stress-related conditions such as celiac disease (CD). CD is a small intestine disorder that results from an inappropriate immune response to gluten in genetically predisposed individuals [16,17,18]. It affects approximately 1% of the global population, though underdiagnosis remains a challenge due to its broad clinical spectrum, with only 1 in every 7 to 10 cases being diagnosed [19]. This abnormal immune response triggers a cascade of reactions leading to intestinal villi damage, impaired nutrient absorption, and a range of secondary complications [20]. The keystone event in CD pathogenesis is the activation of a gluten-specific immune response driven by molecular interactions between gluten, HLA-DQ2/8 (the main genetic predisposition factor), and transglutaminase 2, the CD-specific autoantigen. The innate immune response initiates epithelial damage, while the adaptive response—mediated by CD4+ T cells in the lamina propria—leads to inflammation and intestinal injury. Concurrently, cytotoxic CD8+ intraepithelial lymphocytes become activated, cooperating with cytokines released by gluten-specific CD4+ T cells and interleukin-15 (produced in excess) to orchestrate an autoimmune attack on the gut epithelium [18,21,22,23,24]. In addition to intestinal symptoms, patients often experience extraintestinal manifestations such as anemia, fatigue, and neurological or psychiatric disorders [25,26]. CD is no longer considered a purely pediatric condition, as it can manifest at any age and has been associated with systemic manifestations affecting multiple organs [27,28,29,30]. This is because their appearance and progress are a consequence of interactions between genetic susceptibility and environmental factors [31], including emotional stress and mood disorders such as anxiety [26,27,28,29,32]. Chronic stress and anxiety can lead to increased synthesis of key mediators involved in the stress response. However, prolonged activation of these mediators negatively impacts immune function, promoting chronic oxidative stress and inflammation, which contribute to disease progression [29,33]. Given these findings, experimental models in adult mice with inadequate stress responses and heightened anxiety levels have been explored as potential models for CD susceptibility.

Several years ago, our research group developed a premature aging mouse (PAM) model based on the hyperreactivity of chronologically adult animals to stress. These mice exhibit immunosenescence, increased oxidative stress, premature aging of homeostatic systems, anxiogenic-like responses, and a shorter lifespan compared to their non-aging counterparts with an adaptive stress response [34,35,36,37]. Strikingly, peritoneal leukocytes from PAM exposed to gliadin peptides show immune function alterations similar to those observed in CD patients [17]. Based on these findings and the established relationships between chronic physiological stress, anxiety, and gastrointestinal immune dysfunctions, we hypothesize that PAM may experience further impairments in their immune and redox status following gliadin ingestion. Thus, the present study aimed to evaluate the effects of a gliadin-rich diet over 4 weeks in PAM, focusing on immune function alterations assessed in peritoneal, splenic, and thymic leukocytes, as well as oxidative stress parameters in both immunological and non-immunological organs.

## 2. Materials and Methods

### 2.1. Classification of Prematurely Aging Mice (PAM)

The T-maze consists of three wooden arms, with the interior lined with black methacrylate. The test was performed by placing the mouse in the vertical arm of the maze, facing the wall, and timing how long it took the animal to cross the intersection of the three arms with both hind legs. This test was conducted once a week for four consecutive weeks [36,37]. Mice that took more than 10 s to complete the task on all four test trials were considered PAM.

### 2.2. Animals

Fourteen ex-reproductive Swiss female adult PAM (*Mus musculus*; 32 ± 4 weeks old) were employed for this study. All animals were housed at 5 per cage and placed in the Animal Facility at the Faculty of Biology (Complutense University of Madrid, UCM). The average temperature was 22 ± 2 °C, relative humidity was 60%, and a 12/12 h reversed light/dark cycle (lights on at 20:00 h). These conditions were maintained to avoid circadian interference. Mice were checked daily. Tap water and food were available *ad libitum*.

After acclimatization and PAM classification, mice were divided into two groups of seven individuals each. One group was fed a control diet (A04, Panlab), while the other group was fed a gliadin-rich diet (120 g/kg) for 4 weeks. Afterward, the animals were sacrificed under current regulations (Royal Decree 53/2013), and leukocytes were extracted from the peritoneal cavity along with the organs to be studied: spleen, thymus, lung, heart, kidney, and liver. Mononuclear leukocytes were isolated from the thymus and half of the spleen, and, along with peritoneal leukocytes, immune function assays were performed. Organs and leukocyte cell pellets were stored at −80 °C for subsequent oxidative stress assays.

### 2.3. Diet

Gliadin was provided by Dr. Yolanda Sanz. The gliadin-rich diet was prepared by supplementing the standard control diet with 120 g/kg of gliadin while maintaining the same overall nutritional composition. This approach ensured that any observed effects were specifically due to gliadin inclusion. The selected concentration was based on previous studies, aiming to induce significant changes in immune function and redox balance without reaching excessive levels. Standard rodent diets typically contain about 5% gluten, whereas this formulation provided a gluten-rich diet with 12% gluten, comparable to levels found in human foods such as wheat flour (10–12%). This dietary composition allowed for a relevant evaluation of gliadin’s impact on immune function and oxidative stress, reflecting gluten levels present in certain processed foods consumed by humans [38,39,40].

### 2.4. Blood Measurements

Glucose, cholesterol, and triglyceride levels were quantified from blood drops obtained from the tail vein using reactive strips and an automatic meter (Accutrend Roche, Mannheim, Germany). All analyses were performed according to the manufacturer’s protocols.

### 2.5. Immune Function Parameters

#### 2.5.1. Isolation of Peritoneal, Splenic, and Thymic Leukocytes

The extraction of peritoneal leukocytes carried out before the sacrifice was performed by injecting 2 mL of 37 °C sterile Hank’s solution into the peritoneal cavity of each animal, and subsequent extraction. Splenic and thymic leukocytes were obtained by macerating the specific organ (spleen or thymus) in 3 mL of sterile PBS and centrifuging it in a density gradient using Histopaque 1.077 g/mL at 840× *g* for 20 min at 4 °C. The mononuclear leukocyte layer was collected and washed with sterile PBS. After that, suspensions were centrifuged (400× *g* for 10 min at 4 °C), supernatants were discarded, and cell pellets were re-suspended in 900 μL of sterile PBS.

The cell suspensions obtained were adjusted to the concentration of macrophages, lymphocytes, or total leukocytes specific to the parameter to be analyzed. The cell count was carried out using a Neubauer hemocytometer with the help of the 40× lens of an optical microscope. Cell viability was determined by the blue triptan vital dye exclusion test. Only cell suspensions exceeding 95% cellular viability were used. All incubations were carried out at 37 °C, in a humidified atmosphere and with 5% CO_2_.

#### 2.5.2. Phagocytic Capacity

The phagocytic capacity of peritoneal cells was assessed using a method previously described [36,37]. This method is based on the ability of phagocytic cells to ingest an inert compound (latex beads), reflecting their in vivo phagocytic behavior. 200 μL of peritoneal leukocyte suspensions were adjusted to 0.5 × 10^6^ macrophages/mL and placed in a MIF plate (Sterilin, Teddington, England) for 30 min to generate a monolayer composed of macrophages. These monolayers were then washed, and 200 μL of Hank’s solution and 20 μL of a 1% latex bead suspension (1.091 ± 0.0082 μm in diameter) were added. After 30 min of incubation, the plates were washed, fixed, and stained. The Phagocytic Index (the number of latex particles ingested by 100 macrophages) and Phagocytic Efficiency (percentage of macrophages capable of ingesting at least one latex particle) were determined by optical microscopy (×100).

#### 2.5.3. Chemotaxis Capacity

The chemotactic ability of peritoneal, splenic, and thymic cells was assessed employing the Boyden method (1962) [41], with modifications made by our group [36,37,42]. This method is based on the ability of immune cells to migrate toward an infectious focus, which is mimicked by the presence of a chemotactic agent. The Boyden chamber consists of two compartments separated by a 3 μm-pore nitrocellulose filter. In the upper compartment, 300 μL of leukocyte suspension (adjusted to 0.5 × 10^6^ lymphocytes/mL or 0.5 × 10^6^ macrophages/mL) was placed, while the lower compartment contained 400 μL of the chemotactic agent, the formylated peptide (fMet-Leu-Phe) from *E. coli*, at a concentration of 10^−8^ M. Chambers were incubated for 3 h, filters were fixed and stained. The Chemotaxis Index (C.I.) for each cell type was calculated by counting the number of cells in one-third of the lower face of the filter using optical microscopy (×100).

#### 2.5.4. Natural Killer Activity

The Natural Killer (NK) activity of peritoneal, splenic, and thymic cells was assessed using a previously described method [36,37,42,43], based on a colorimetric measurement of target cell lysis by determining the enzyme lactate dehydrogenase (LDH) activity, using tetrazolium salts (Cytotox 96™ kit, Promega, Madison, WI, USA). This analysis reflects the ability of certain immune cells (NK cells) to kill tumor cells. Briefly, 100 μL aliquots of target cells (YAC-1 murine lymphoma cells) at a concentration of 10^4^ cells/well were seeded in U-bottom 96-well plates. Subsequently, 100 μL aliquots of effector cell suspensions (peritoneal, splenic, and thymic leukocytes, respectively) at a concentration of 10^5^ cells/well were added, achieving an effector/target ratio of 10/1. Both cell suspensions were adjusted in RPMI-1640 medium, lacking phenol red pH indicator to avoid potential interference with the subsequent colorimetric analysis. Plates were incubated for 4 h. After incubation, 50 μL of supernatants from each well were collected to measure absorbance at 490 nm for LDH activity by adding the enzyme substrate. The results were expressed as the percentage of target cell lysis. The calculation was performed using the following equation:$$\%\;\mathrm{lysis}=\frac{\mathrm{Effector}\;\mathrm{cells}\;\mathrm{lysis}-\mathrm{spontaneous}\;\mathrm{lysis}\;\mathrm{of}\;\mathrm{the}\;\mathrm{effector}\;\mathrm{cells}-\mathrm{spontaneous}\;\mathrm{lysis}\;\mathrm{of}\;\mathrm{the}\;\mathrm{tumoral}\;\mathrm{cells}}{\mathrm{Total}\;\mathrm{lysis}\;\mathrm{of}\;\mathrm{the}\;\mathrm{tumoral}\;\mathrm{cells}-\mathrm{spontaneous}\;\mathrm{lysis}\;\mathrm{of}\;\mathrm{the}\;\mathrm{tumoral}\;\mathrm{cells}}\times 100$$

Effector cell lysis refers to the average absorbance of the wells, where the observed lysis is caused by the action of the effector cells (peritoneal, splenic, and thymic lymphocytes, respectively) on the target cells (YAC-1). Spontaneous lysis of effector cells is the lysis caused by the natural death of peritoneal, splenic, and thymic leukocytes, respectively, during the process. Total target cell lysis is the average absorbance of wells, where all tumor cells (YAC1 cells) have been lysed by the addition of a lysis solution. Finally, spontaneous lysis of tumor cells is the average absorbance of the lysis caused by the natural death of tumor cells during the process.

#### 2.5.5. Lymphoproliferative Capacity

The proliferative capacity of peritoneal lymphocytes was assessed following a previously described method [35,37]. Lymphoproliferation was evaluated both at the basal level and under stimulation with two different mitogens: concanavalin A (ConA) and lipopolysaccharide (LPS). This test measures the proliferative response to mitogens in vitro, mimicking the response to antigens that occur in vivo. The method is based on the ability of mature lymphocytes to transform into dividing cells or lymphoblasts under appropriate conditions. The assay takes advantage of this division capacity by incorporating tritiated thymidine into the DNA being synthesized, as a synthetic analog. Aliquots of 200 μL of the leukocyte suspensions (adjusted to 10^6^ lymphocytes/mL) were cultured in flat-bottom 96-well plates in complete medium (RPMI-1640 supplemented with 10% heat-inactivated fetal bovine serum and 1% gentamicin) for 48 h, either in the absence (basal lymphoproliferation) or presence of the mitogens ConA (1 μg/mL per well) or LPS (1 μg/mL per well). After this period, 5 μL of tritiated thymidine (2.5 μCi/mL) was added to each well, and the complete medium was renewed. After 24 h of incubation, cells were collected using a semi-automatic harvester, and the thymidine incorporated by the lymphocytes was measured in a beta counter. The results were expressed in counts per minute (c.p.m).

### 2.6. Oxidative Stress Parameters

#### 2.6.1. Glutathione Reductase (GR) Activity

Glutathione reductase (GR) activity was measured by spectrophotometry following the protocol described by Massey and Williams (1965) [44], with slight modifications [37,42,43]. For the measurement, tissue samples were homogenized in a 50 mM phosphate buffer with 6.3 mM EDTA, pH 7.4 (previously helium-bubbled). In the case of peritoneal, splenic, and thymic leukocytes, these samples were sonicated in 200 µL of buffer. Then, all samples were centrifuged, and supernatants were collected to determine GR activity. The results were expressed as milliunits (mU) per milligram of protein in the case of organs, and the activity in cells was expressed as a mU per million cells.

#### 2.6.2. Glutathione Peroxidase (GPx) Activity

The activity of glutathione peroxidase (GPx) was measured by spectrophotometry following the protocol described by Lawrence and Burk (1976) [45], with slight modifications [39,40]. For the measurement, tissue samples were homogenized, and cells were sonicated in a 50 mM phosphate buffer, pH 7.4 (previously helium-bubbled), and the samples were centrifuged at 4 °C. The supernatants were then collected, and the GPx activity was determined in them. The results were expressed as a mU per milligram of protein for the organs, and mU per million cells for cell samples.

#### 2.6.3. Oxidized (GSSG) and Reduced (GSH) Glutathione Concentrations

The determination of glutathione concentrations in samples was performed using the modified method of Hissin and Hilf (1976) and adapted for plate analysis by using the fluorescent probe O-phthalaldehyde (OPT) [37,42,43]. This method is based on the ability of the probe to bind to GSH at pH 8 and GSSG at pH 12. For the measurement, tissue samples were homogenized in phosphate buffer containing EDTA (0.1 M, pH 8). In the case of peritoneal, splenic, and thymic leukocytes, these samples were sonicated in 200 µL of buffer. Then, all samples were centrifuged, and supernatants were collected to determine GSSG and GSH concentrations. The results were expressed as nmol of GSSG or GSH per milligram of protein for organs, and per million cells for thymic and splenic cells. Also, GSSG/GSH ratios were calculated.

#### 2.6.4. Xanthine Oxidase (XO) Enzymatic Activity

The analysis of XO activity was performed using a fluorometric assay with the commercial kit “Amplex Red Xanthine/Xanthine Oxidase Assay Kit” (Invitrogen, Eugene, OR, USA) [46]. For the measurement, tissue samples were homogenized in the assay buffer. In the case of peritoneal, splenic, and thymic leukocytes, these samples were sonicated in 200 µL of buffer. The results were expressed in milli-international units (mU) of enzymatic activity per milligram of protein for organs. For splenic, thymic, and peritoneal cells, the results were expressed as mU of enzymatic activity per million cells.

#### 2.6.5. Lipid Peroxidation: TBARS Concentration

Determination of TBARS concentration was performed using the commercial kit “Lipid Peroxidation (TBARS) Assay Kit” (BioVision, Milpitas, CA, USA), as previously described with some modifications [36,37,42,43,47]. For the measurement, tissue samples were homogenized in a lysis buffer containing butylated hydroxytoluene (BHT) (0.1 mM) to prevent further TBARS formation during the process. In the case of peritoneal, splenic, and thymic leukocytes, these samples were sonicated in 200 µL of lysis buffer. The supernatants were incubated with thiobarbituric acid (TBA) for 60 min in a water bath at 95 °C. Later, pure butanol was added. Samples were centrifuged and the organic phase was dispensed into 96-well plates for spectrophotometric measurement at 532 nm. The results were expressed as nmol of TBARS/mg protein.

### 2.7. Protein Concentration

This assessment was carried out using the bicinchoninic acid (BCA) test with the BCA kit, which is based on the reduction in Cu^2+^, generating Cu^+^ ions that bind to the BCA and form a colored compound analyzed by spectrophotometry at 562 nm [36,42,43,47].

### 2.8. Statistical Analysis

The data were expressed as a mean ± standard deviation and were analyzed using SPSS 21.0 (SPSS Inc., Chicago, IL, USA). In all cases, the normal distribution was first checked using the Kolmogorov–Smirnov test, and the homogeneity of variances using the Levene test. A media comparison was then made using a non-paired Student *t*-test for the blood measurements and immune and oxidative parameters, while the differences in food and water intake and body weight were studied using a paired Student *t*-test. Significant differences were considered to exist in cases where the *p*-value was lower than 0.05. Thus, *p*-value = 0.05 was considered significant, *p* = 0.01 was very significant, and *p* = 0.001 was highly significant.

## 3. Results

The results showed no significant differences in body weight (Figure 1C) or food and water intake (Figure 1B,D) between mice fed with a gliadin-rich diet and those on a standard diet. However, mice consuming a gliadin-rich diet exhibited a trend toward higherserum cholesterol levels (Figure 1E, *p* = 0.06) and significantly elevated blood glucose levels (Figure 1F, *p* = 0.001) compared to the control group. No significant dietary differences in triglyceride levels were obtained (Control: 143.17 ± 14.19 and Diet: 143 ± 9.2 mg/mL).

Regarding the immune function parameters analyzed (Figure 2), the results show that mice fed a gliadin-rich diet exhibited alterations, with lower phagocytic efficiency (Figure 2A, *p* = 0.001) and natural killer (NK) (Figure 2B, *p* = 0.001) activity in peritoneal, splenic (Figure 2G), and thymic (Figure 2H) leukocytes than the control group. Conversely, these mice exhibited higher chemotactic activity in peritoneal (Figure 2C, *p* = 0.003) and splenic lymphocytes (Figure 2F) and lymphoproliferative responses to Concanavalin A (ConA) in peritoneal leukocytes (Figure 2D, *p* = 0.03) than those fed a control diet. Additionally, peritoneal leukocytes from PAM fed a gliadin-rich diet showed a higher basal lymphoproliferation, an indirect marker of inflammation, than control PAM (Figure 2E, *p* = 0.054). No statistical differences in peritoneal cells were obtained for phagocytic index (Control: 182 ± 28, Diet: 167 ± 24), macrophage chemotaxis (Control: 29 ± 17, Diet: 33 ± 18), LPS-stimulated lymphoproliferation (c.p.m) (Control: 1611 ± 179, Diet: 1997 ± 162.7), and in the chemotaxis of thymic lymphocytes (Control: 350 ± 120, Diet: 280 ± 100).

The oxidative stress parameters analyzed in splenic and thymic leukocytes (Figure 3) show that mice fed with a gliadin-enriched diet present lower antioxidant glutathione peroxidase (GPx) (*p* = 0.000) and glutathione reductase (GR) (*p* = 0.031) activities in thymic cells, and lower reduced glutathione (GSH) concentration in splenic and thymic cells (*p* = 0.003 and *p* = 0.03, respectively) than the control group. In addition, these mice also present higher oxidized glutathione (GSSG) concentration in thymic (*p* = 0.000) and splenic (*p* < 0.009) leukocytes, than in the control group. Also, they present a higher GSSG/GSH ratio in thymic (Control: 0.87 ± 0.15, Diet: 8.3 ± 2.3) and splenic leukocytes (Control: 0.27 ± 0.015, Diet: 3.66 ± 1.2) with a *p*-value = 0.000 and *p*-value = 0.033, respectively.

In Figure 4, we can see the results obtained for the oxidative state in several organs (liver, lung, kidney, spleen, and heart). The activity of the enzyme glutathione peroxidase (GPx) is shown in Figure 4A. It can be observed that the activity in all the organs studied was significantly lower in the DIET group than in the control group. The significant differences were *p* = 0.017 (liver), *p* = 0.005 (lung), *p* = 0.013 (kidney), *p* = 0.006 (spleen), and *p* = 0.004 (heart). The same was observed in the results on the enzymatic activity of glutathione reductase (GR) (Figure 4B), with the statistical differences of *p* = 0.005 (liver), *p* = 0.02 (lung), *p* = 0.029 (kidney), *p* = 0.050 (cortex) and *p* = 0.005 (spleen). For the GSH concentrations (Figure 4C), these were lower in the liver (*p* = 0.017), lung (0.005), and kidney (*p* = 0.004) of mice fed a gliadin-rich diet than in the control group. A significant trend was observed in the spleen (*p* = 0.065) and the heart (*p* = 0.06). Figure 4D shows that PAM fed with a diet rich in gliadin have a higher concentration of GSSG in the liver (*p* = 0.002), kidney (*p* = 0.004), and spleen (0.025), while no difference has been observed in the lung and heart. They also had a higher enzymatic activity of xanthine oxidase (Figure 4E) in the lung (*p* = 0.035), spleen (*p* = 0.027), and heart (*p* = 0.007), as well as a higher concentration of TBARS (Figure 4F) in the lung (*p* = 0.001) and the heart (*p* = 0.057), and a significant trend in the spleen (*p* = 0.082).

## 4. Discussion

This is the first study to describe how PAM fed with a gliadin-rich diet worsen their immune and redox impairments. These animals, after 4 weeks of gliadin diet ingestion, exhibit a general impairment of their immunity, exhibiting an oxidative stress establishment not only in immune locations but also in extra-immune organs. Our results demonstrate that a gliadin-rich diet significantly affects multiple immune functions in leukocytes derived from the peritoneum, spleen, and thymus from prematurely aging mice, emphasizing the wide-ranging immunomodulatory effects of gliadin. Specifically, the observed lower phagocytic efficacy of peritoneal macrophages and the diminished NK cell activity in all leukocyte samples, compared to controls, highlights gliadin’s detrimental impact on innate immunity. These functions, which are essential for the early defense against infections, are also further impaired in celiac disease (CD), a condition often associated with viral infections during onset [48,49]. The reduced activity of NK cells, characterized by a diminished capacity to recognize and eliminate virus-infected cells, is exacerbated by a decreased presence of critical receptors on these cells in CD patients, further reinforcing the disease’s immunological vulnerability [7].

Previous studies have demonstrated that PAM exhibit lower baseline levels of phagocytic capacity and NK cell activity compared to their non-prematurely aging counterparts (NPAM) [36,37,42]. The gliadin-rich diet intensifies this decline, suggesting that gluten not only accelerates immune aging but also compromises the body’s ability to mount an effective innate immune response. Gliadin is a main component of gluten, a compound closely linked with gastrointestinal diseases, such as CD [1,2,3,4]. Considering this relation, these findings could be particularly relevant for CD patients, who already experience systemic immune dysregulation that extends beyond the gastrointestinal tract. Indeed, the link between compromised NK cell function and an increased prevalence of malignancy in CD [50,51] could align with the hypothesis that gliadin may act as a carcinogen, creating a pro-inflammatory and immunosuppressive environment, that triggers tumor development [7]. In fact, an association between CD and cancer has been proposed [51]. In addition to its effects on innate immunity, gliadin appears to significantly influence adaptive immune responses. This study found a marked increase in lymphocyte chemotaxis and lymphoproliferation in response to ConA in PAM fed a gliadin-rich diet (PAMD). This aligns with the inflammatory profile of CD, which is driven by CD4 T lymphocytes that migrate to the site of injury to eliminate antigens. Gliadin’s ability to induce immune cell migration, akin to the chemotactic action of formylated peptides such as fMet-Leu-Phe from E. coli [8], highlights its role as a potent immune stimulus. However, this heightened chemotaxis and lymphoproliferation could exacerbate tissue damage and chronic inflammation in CD, further perpetuating the disease’s pathological cycle. Taken together, these findings underscore the systemic nature of immune abnormalities in CD, where the effects of gliadin extend beyond the gut to impact immune cells in other locations. The interplay between impaired innate immunity-heightened adaptive immune activity and chronic inflammation creates a dynamic that not only facilitates disease progression but also increases the risk of malignancy. Understanding the mechanisms by which gliadin influences these immune functions may provide new insights into the pathogenesis of gastrointestinal diseases, such as CD, highlighting potential therapeutic targets to mitigate its systemic impact.

The presented results also confirm that a gliadin-rich diet negatively impacts the antioxidant defenses of PAM, specifically the enzymatic activity of glutathione reductase and glutathione peroxidase, as well as the amount of reduced glutathione (GSH), although this is not observed in all tissues or cell types. The decreased activity of antioxidant enzymes seems to indicate that PAM, which already have a worse redox state compared to NPAM [36,37], may worsen their redox condition upon consuming a gliadin-containing diet. The liver and kidneys are the main organs involved in the synthesis and degradation of GSH, as well as in inter-organ circulation [52]. In this regard, the results are consistent, showing a significant decrease in GSH in both the liver and kidneys of PAMD, as well as in the lungs. In addition, the results show that the gliadin-rich diet also affects oxidants, generally increasing them, as seen in the amount of oxidized glutathione (GSSG), the enzymatic activity of xanthine oxidase, and lipid peroxidation (TBARS concentrations). Therefore, the gliadin-rich diet not only decreases antioxidant defenses in PAM but also increases oxidative components. This increase in oxidant compounds generated by a gliadin rich diet affects PAM that already have an oxidative stress establishment due to premature aging [36,37]. In fact, this oxidative stress is patent in the GSSG/GSH ratio evaluated in splenic and thymic leukocytes from PAM fed with a gliadin-rich diet. The GSSG/GSH ratio is a sensitive indicator of changes in a cell’s thiol redox state and ongoing redox signaling, as well as the functional state of the cell [53]. Regarding lipid damage, it seems that the gliadin-rich diet did not affect lipid peroxidation in cells. However, in organs, the effect of the gliadin-rich diet on lipid peroxidation can be observed, specifically in the lungs, spleen, and heart, with higher concentrations of TBARS in the PAM diet group compared to the controls. This suggests that the gliadin-rich diet increased the oxidative degradation of the cell membrane. In the spleen, the diet led to the lower enzymatic activity of both glutathione peroxidase and glutathione reductase, as previously discussed, along with an increase in xanthine oxidase activity. However, no lower values of GSH in PAMD were observed; instead, both GSH and GSSG levels were higher. These results suggest that in this organ, the body is attempting to compensate for the oxidative stress exacerbated by the gliadin-rich diet. Thus, the increase in GSH in PAMD would demonstrate an attempt to compensate for oxidative stress, where the liver’s GSH synthesis capacity could be involved. The liver is the organ with the highest GSH concentrations, being central to the biosynthesis of this antioxidant, and its high GSH concentration is related to the liver’s role in detoxification and xenobiotic elimination [52]. It could be suggested that the GSH generated in the liver is transported to the spleen, an organ with a significant immune component, where antigen-presenting cells (APCs) are large glutathione captors [52]. Therefore, in the spleen, the organism is attempting to compensate for the oxidative stress caused by the gliadin-rich diet, but this compensation fails due to a significant increase in oxidative components such as oxidized xanthine or GSSG, and a decrease in the enzymatic activity of glutathione peroxidase and reductase (antioxidant defense enzymes). Indeed, it is important to highlight the broad spectrum of alterations that PAM exhibit after ingesting a gliadin-rich diet, impairments resembling the extra-intestinal alterations previously described in CD patients [7,8,10,25].

An additional and important dimension to this work is the relationship between an inadequate response to stress and the predisposition to develop gastrointestinal pathologies, such as CD. Stress is a well-documented modulator of immune function [54,55,56,57,58,59], and individuals with altered emotional stress behaviors and anxiety often exhibit compromised immune defenses [54,55,56,57,58,59,60,61,62,63]. Indeed, CD has been closely related to the development of psychological symptoms, such as anxiety and depression [26]. Based on that, it is plausible that individuals with an altered emotional stress response could develop or at least exhibit the development of gastrointestinal diseases, such as CD. Notably, PAM also exhibit pronounced hyperreactivity to novel environments and elevated anxiety levels [33,34,35]. Previous research by De Palma et al. (2014) [17] demonstrated that peritoneal leukocytes from prematurely aging mice (PAM) show impaired in vitro responses to gliadin and gut microbes. These impairments are characterized by reduced phagocytic activity, heightened inflammation, and altered cytokine profiles. These findings resemble those obtained here in PAM exposed to gliadin ingestion and indicate the establishment of an inflammatory and oxidative stress consequence of gliadin-rich diet ingestion. This presence of oxidative stress in PAM parallels the pathological features of CD, suggesting that individuals with stress-related immune impairments and heightened oxidative stress may be particularly susceptible to developing this disease. Additionally, previous reports have indicated that PAM display a clear establishment of oxidation–inflammation processes in both immune and non-immune tissues at basal state [36,37,42,64], which are hallmark features not only of aging but also of celiac disease (CD) [16,33,65,66,67]. Taken together, these findings support a link between altered stress responses and the negative impact of high-gluten diets. Specifically, mice exhibiting stress-related behavioral changes and premature immune senescence, such as PAM [35], might serve as valuable models for studying the negative impact of gluten on immunity and the oxidative state, and consequently on the health of individuals with inadequate stress responses. This is particularly relevant in today’s society, where elevated stress levels and increased cases of stress and anxiety are prevalent. Understanding the interplay between stress, gliadin exposure, and immune dysregulation offers important insights into how these factors affect health and could lead to potential therapeutic targets.

It is important to highlight some limitations of this study. First, although celiac disease (CD) has a higher incidence in women than in men [68,69,70], and for this reason we used female PAM in this work, future research should also include male mice with premature aging. This would help determine whether these animals, when fed a gliadin-rich diet, exhibit similar impairments to those observed in female PAM. Second, while our study focused on evaluating immune function parameters and oxidative stress markers, we recognize that key features of CD need to be further explored to better validate our model. Future experiments will incorporate laboratory tests to assess localized inflammation and immune cell activation in the small intestine, intestinal villi deterioration, cytoarchitecture, and epithelial integrity, as well as immune cell infiltration analysis. Moreover, the assessment of specific CD markers, such as anti-tissue transglutaminase, anti-endomysial, and anti-deamidated gliadin peptide antibodies [71,72] or serum anti-actin antibodies [73] will be also included in future investigations to further strengthen the translational relevance of our model.

## 5. Conclusions

Prematurely aging mice (PAM), animals that exhibit an inadequate stress response and high anxiety levels, present immune function alterations after ingesting a gliadin-rich diet. While activation of various T lymphocyte functions was observed, there was a reduction in relevant innate immune functions similar to those seen in patients with CD. This imbalance in immunity in adult PAM following gliadin-rich diet ingestion, particularly impairments related to the innate immune response, highlights the potential immunosenescence induced by this dietary component. Furthermore, these PAM exhibited signs of oxidative stress affecting both immune and non-immune organs, resembling extra-intestinal alterations described by other studies. Given that, and considering the negative consequences of a gliadin-rich diet observed here and in previous studies with CD patients, these mice might be useful for studying the broader effects of gluten on health and developing strategies targeting immunity and oxidative stress.

## Figures and Tables

**Figure 1 cells-14-00279-f001:**
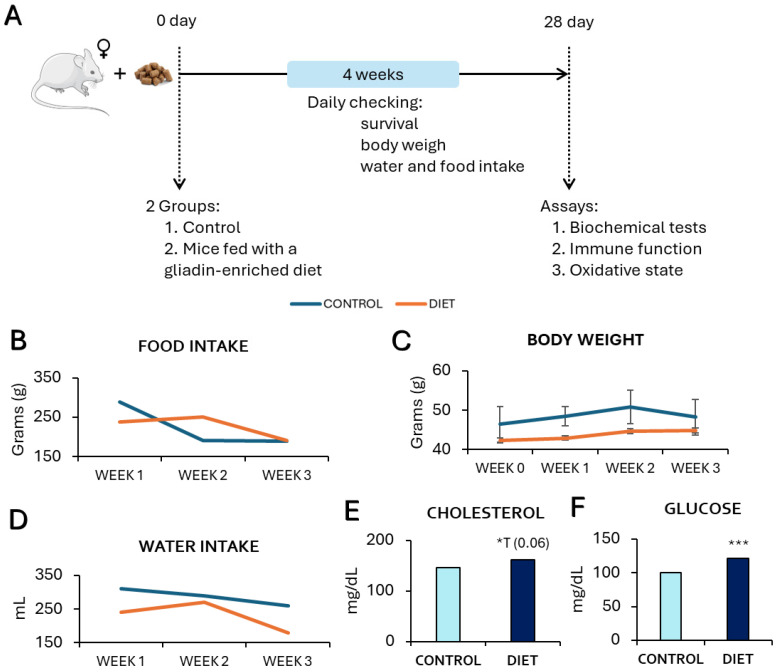
Effects of a gliadin-enriched diet on food and water intake, body weight, and cholesterol and glucose concentrations from PAM. (**A**) Graphical summary of the experimental design; (**B**) Food intake in grams (g); (**C**) Body weight in grams (g); (**D**) Water intake in mL; (**E**) Cholesterol concentration in mg/mL; (**F**) Glucose concentration in mg/dL. Each column shows as mean ± standard deviation. The statistical analysis used was non-paired t-student for cholesterol and glucose and paired t-student for food and water intake and body weight changes *: *p* < 0.05 and *** *p* < 0.001 concerning values obtained in the control group. *T: statistical trend.

**Figure 2 cells-14-00279-f002:**
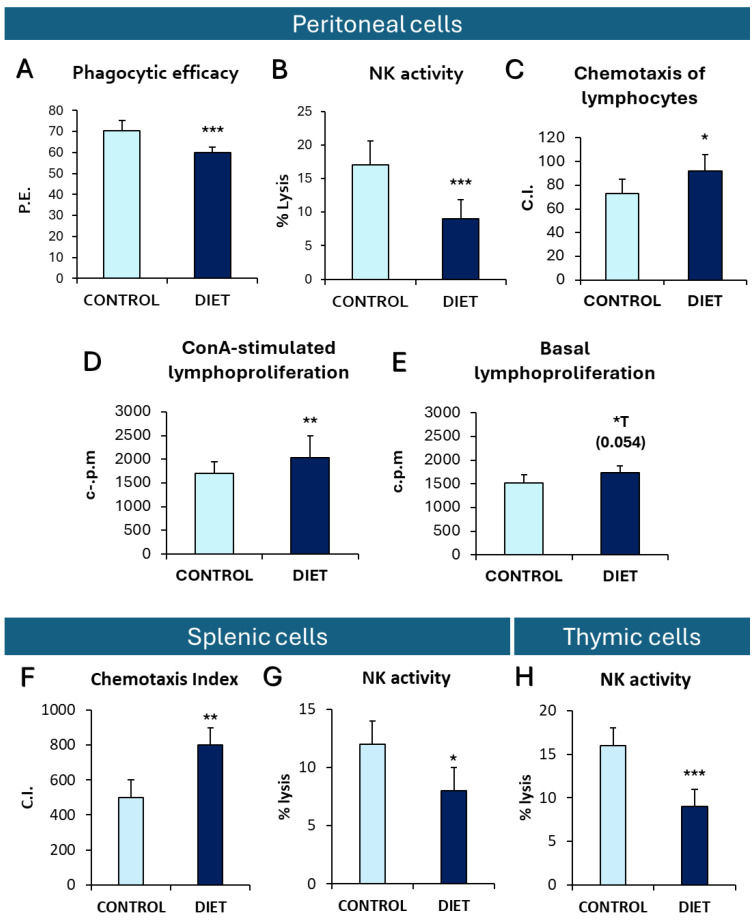
Immune function parameters. In peritoneal cells: (**A**) Phagocytic efficacy (P.E.), (**B**) Natural killer (NK) activity (% lysis), (**C**) chemotaxis index (C.I.) of lymphocytes, (**D**) ConA-stimulated lymphoproliferation (counts per minute or c.p.m) and (**E**) Basal lymphoproliferation (c.p.m). In spleen leukocytes: (**F**) chemotaxis index (C.I.) and (**G**) NK activity (% lysis). In thymic leukocytes: (**H**) NK activity (% lysis). Each column shows as mean ± standard deviation. The statistical analysis used was unpaired t-student *: *p* < 0.05; **: *p* < 0.01 and *** *p* < 0.001 with respect to the values obtained in control animals. *T: statistical trend.

**Figure 3 cells-14-00279-f003:**
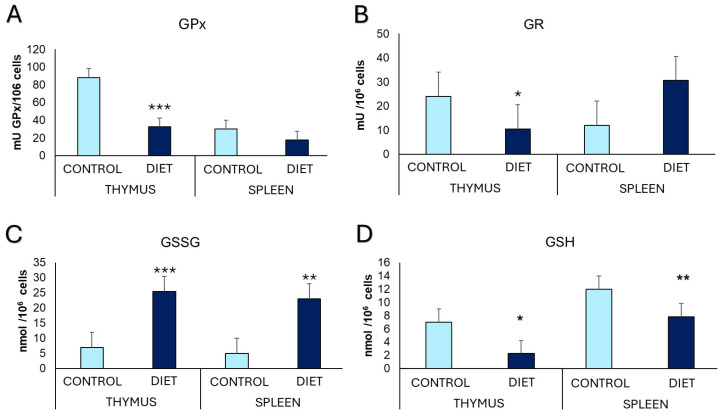
Oxidative state parameters evaluated in splenic and thymic leukocytes. (**A**) Glutathione peroxidase (GPx) activity in mU/10^6^ cells, (**B**) Glutathione reductase (GR) activity in mU/10^6^ cells, (**C**) Oxidized glutathione (GSSG) in nmol/10^6^ cells, and (**D**) Reduced glutathione (GSH) in nmol/10^6^ cells. Each column shows as mean ± standard deviation. The statistical analysis used was unpaired t-student *: *p* < 0.05; **: *p* < 0.01 and *** *p* < 0.001 concerning the values obtained in the control group.

**Figure 4 cells-14-00279-f004:**
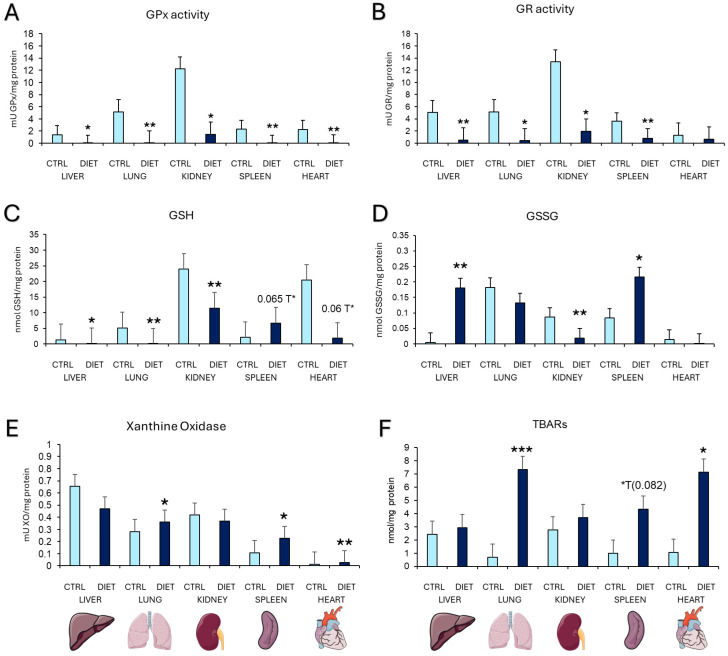
Oxidative state parameters in several organs: liver, lung, kidney, spleen, and heart. (**A**) Glutathione peroxidase (GPx) activity in mU/mg protein, (**B**) Glutathione reductase (GR) activity in mU/mg protein, (**C**) Glutathione reduced (GSH) in nmol/mg protein, (**D**) Glutathione oxidized (GSSG) in nmol/mg protein, (**E**) Xanthine oxidase activity in mU/mg protein, and (**F**) TBARS concentration in nmol/mg protein. Each column shows as mean ± standard deviation. The statistical analysis used was the t-student *: *p* < 0.05; **: *p* < 0.01 and *** *p* < 0.001 concerning the values obtained in the control group. *T: statistical trend.

## Data Availability

Data will be provided if required.

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
