# Peer review of "Gliadin-Rich Diet Worsens Immune and Redox Impairments in Prematurely Aging Mice"

_cells, 2025, doi:10.3390/cells14040279_

Round 1

Reviewer 1 Report

Comments and Suggestions for Authors

This study aimed to analyze the effects of a gliadin-rich diet over 4 weeks in  prematurely aging mice (PAM), focusing on a series of immune functions evaluated by both peritoneal, splenic, and thymic leukocytes and oxidative stress parameters analyzed in immunological and non-immunological organs. Fourteen ex-reproductive Swiss female adult PAM were employed for this study. After acclimatization and PAM classification, mice were divided into two groups: adult female PAM fed a gluten-enriched diet (PAMD, 120 g/kg) or a standard diet (PAMC) for four weeks. Immune function parameters in peritoneal, splenic, and thymic leukocytes (phagocytosis, chemotaxis, Natural Killer activity, lymphoproliferation) and redox markers (glutathione reductase, glutathione peroxidase, reduced/oxidized glutathione, xanthine oxidase activity, lipid peroxidation) were assessed. Then, the animals were sacrificed and leukocytes were extracted from the peritoneal cavity along with the organs to be studied: spleen, thymus, lung, heart, kidney, and liver. Mononuclear leukocytes were isolated from the thymus and half of the spleen, and, along with peritoneal leukocytes, immune function assays were performed. They found that mice consuming a gliadin-rich diet exhibited a trend toward higher serum cholesterol levels and significantly elevated blood glucose levels compared to the control group. Mice fed a gliadin-rich diet exhibited altered, with lower phagocytic efficiency and natural killer activity in peritoneal, splenic, and thymic leukocytes than the control group. Conversely, these mice exhibited higher chemotactic activity in peritoneal and splenic lymphocytes and lymphoproliferative responses to Concanavalin A (ConA) in peritoneal leukocytes than those fed with a control diet. Thus,  PAMD displayed more impaired immune function, lower antioxidant enzyme activities, reduced glutathione, and higher oxidized glutathione concentrations and xanthine oxidase activity than PAMC. The authors concluded that their results indicate that PAM fed with a gliadin-rich diet worsen their immune and redox impairments, thus supporting PAM as a valuable in vivo model for celiac disease.

The study is of interest, however, some issues shoudl be addressed to support the study conclusions. In my opinion, to validate the reliability of an animal model it would be mandatory to verify the development of pathological features resembling the original disease changes. Here the authors demonstrate that a gliadin-rich diet significantly affects multiple immune functions in leukocytes derived from the peritoneum, spleen, and thymus from prematurely aging mice, emphasizing the wide-ranging immunomodulatory effects of gliadin. However, whether these immune function changes translate into intestinal mucosal damage has not verified. The authors should suggest how this animal model could be improved by adding laboratory tests assessing an indirect measure of mucosal damage. In particular, it is well known that intestinal mucosal damage in celiac disease is characterized by an increased intestinal permeability. More importantly, in CD patients with villous atrophy, the development of serum anti-actin antibodies is a high CD-specific immunological features as previously demonstrated (Clin Exp Immunol. 2004 Aug;137(2):386-92. doi: 10.1111/j.1365-2249.2004.02541.x. ). Therefore, they could consider in future studies to evaluate in their novel animal model how intestinal permeability changes over time after starting gliadin-rich diet, as well as to assess the development of anti-actin antibodies in those developing villous atrophy (Clin Exp Immunol. 2004 Aug;137(2):386-92. doi: 10.1111/j.1365-2249.2004.02541.x. ).

Author Response

Comments and Suggestions for Authors

This study aimed to analyze the effects of a gliadin-rich diet over 4 weeks in prematurely aging mice (PAM), focusing on a series of immune functions evaluated by both peritoneal, splenic, and thymic leukocytes and oxidative stress parameters analyzed in immunological and non-immunological organs. Fourteen ex-reproductive Swiss female adult PAM were employed for this study. After acclimatization and PAM classification, mice were divided into two groups: adult female PAM fed a gluten-enriched diet (PAMD, 120 g/kg) or a standard diet (PAMC) for four weeks. Immune function parameters in peritoneal, splenic, and thymic leukocytes (phagocytosis, chemotaxis, Natural Killer activity, lymphoproliferation) and redox markers (glutathione reductase, glutathione peroxidase, reduced/oxidized glutathione, xanthine oxidase activity, lipid peroxidation) were assessed. Then, the animals were sacrificed, and leukocytes were extracted from the peritoneal cavity along with the organs to be studied: spleen, thymus, lung, heart, kidney, and liver. Mononuclear leukocytes were isolated from the thymus and half of the spleen, and, along with peritoneal leukocytes, immune function assays were performed. They found that mice consuming a gliadin-rich diet exhibited a trend toward higher serum cholesterol levels and significantly elevated blood glucose levels compared to the control group. Mice fed a gliadin-rich diet exhibited altered, with lower phagocytic efficiency and natural killer activity in peritoneal, splenic, and thymic leukocytes than the control group. Conversely, these mice exhibited higher chemotactic activity in peritoneal and splenic lymphocytes and lymphoproliferative responses to Concanavalin A (ConA) in peritoneal leukocytes than those fed with a control diet. Thus, PAMD displayed more impaired immune function, lower antioxidant enzyme activities, reduced glutathione, and higher oxidized glutathione concentrations and xanthine oxidase activity than PAMC. The authors concluded that their results indicate that PAM fed with a gliadin-rich diet worsen their immune and redox impairments, thus supporting PAM as a valuable in vivo model for celiac disease.

The study is of interest, however, some issues should be addressed to support the study conclusions. In my opinion, to validate the reliability of an animal model it would be mandatory to verify the development of pathological features resembling the original disease changes. Here the authors demonstrate that a gliadin-rich diet significantly affects multiple immune functions in leukocytes derived from the peritoneum, spleen, and thymus from prematurely aging mice, emphasizing the wide-ranging immunomodulatory effects of gliadin. However, whether these immune function changes translate into intestinal mucosal damage has not verified. The authors should suggest how this animal model could be improved by adding laboratory tests assessing an indirect measure of mucosal damage. In particular, it is well known that intestinal mucosal damage in celiac disease is characterized by an increased intestinal permeability. More importantly, in CD patients with villous atrophy, the development of serum anti-actin antibodies is a high CD-specific immunological feature as previously demonstrated (Clin Exp Immunol. 2004 Aug;137(2):386-92. doi: 10.1111/j.1365-2249.2004.02541.x.). Therefore, they could consider in future studies to evaluate in their novel animal model how intestinal permeability changes over time after starting the gliadin-rich diet, as well as to assess the development of anti-actin antibodies in those developing villous atrophy (Clin Exp Immunol. 2004 Aug;137(2):386-92. doi: 10.1111/j.1365-2249.2004.02541.x.).

Thank you for your constructive feedback and insightful comments regarding our study. We appreciate your attention to detail and the suggestion to assess the development of intestinal mucosal damage in our animal model. We fully agree that the presence of pathological features resembling celiac disease (CD), such as mucosal damage and increased intestinal permeability, is crucial for validating the model as a relevant tool for studying gluten-induced immune dysregulation.

While our study primarily focused on evaluating immune function parameters and oxidative stress markers after gliadin rich diet ingestion in PAM, we recognize that intestinal mucosal damage is a key hallmark of CD, and its assessment would add valuable depth to the current findings. We plan to address this aspect in future studies by incorporating laboratory tests to evaluate these possible impairments. Among them, the analysis of intestinal cytoarchitecture through histopathological studies to check the epithelium integrity as well as the evaluation of immune cell infiltration together with the determination of specific celiac disease markers, such as anti-tissue transglutaminase, anti-endomysial and anti-deamidated gliadin peptide antibodies [72,73] will be considered. We also appreciate your suggestion to evaluate the development of serum anti-actin antibodies, a specific immunological marker for CD, which will also be considered in this subsequent investigation to further strengthen the translational relevance of our model. Following the suggestion of the reviewer, these future analyses have been also included in the revised manuscript, as he/she can check in that and below:

“It is important to highlight some limitations of this study. First, although celiac disease (CD) has a higher incidence in women than in men [69-71], and for this reason, we used female PAM in this work, future research should also include male mice with premature aging. This would help determine whether these animals when fed a glia-din-rich diet, exhibit similar impairments to those observed in female PAM. Second, while our study focused on evaluating immune function parameters and oxidative stress markers, we recognize that key features of CD need to be further explored to better validate our model. Future experiments will incorporate laboratory tests to assess localized inflammation and immune cell activation in the small intestine, intestinal villi deterioration, cytoarchitecture, and epithelial integrity, as well as immune cell infiltration analysis. Moreover, the assessment of specific CD markers, such as an-ti-tissue transglutaminase, anti-endomysial, and anti-deamidated gliadin peptide antibodies [72,73] or serum anti-actin antibodies [74] will be also included in future investigations to further strengthen the translational relevance of our model.”

[69] Ivarsson A, Persson LA, Juto P, Peltonen M, Suhr O, Hernell O. High prevalence of un-diagnosed coeliac disease in adults: a Swedish population-based study. J Intern Med. 1999 Jan;245(1):63-8. doi: 10.1046/j.1365-2796.1999.00403.x.

[70] Gomez JC, Selvaggio GS, Viola M, Pizarro B, la Motta G, de Barrio S, Castelletto R, Echeverría R, Sugai E, Vazquez H, Mauriño E, Bai JC. Prevalence of celiac disease in Argentina: screening of an adult population in the La Plata area. Am J Gastroenterol. 2001 Sep;96(9):2700-4. doi: 10.1111/j.1572-0241.2001.04124.x.

[71] Rutz R, Ritzler E, Fierz W, Herzog D. Prevalence of asymptomatic celiac disease in ado-lescents of eastern Switzerland. Swiss Med Wkly. 2002 Jan 26;132(3-4):43-7. doi: 10.4414/smw.2002.09793.

[72] Singh A, Pramanik A, Acharya P, Makharia GK. Non-Invasive Biomarkers for Celiac Disease. J Clin Med. 2019 Jun 21;8(6):885. doi: 10.3390/jcm8060885.

[73] Ramírez-Sánchez AD, Tan IL, Gonera-de Jong BC, Visschedijk MC, Jonkers I, Withoff S. Mo-lecular Biomarkers for Celiac Disease: Past, Present and Future. Int J Mol Sci. 2020 Nov 12;21(22):8528. doi: 10.3390/ijms21228528.

[74] Granito A, Muratori P, Cassani F, Pappas G, Muratori L, Agostinelli D, Veronesi L, Bortolotti R, Petrolini N, Bianchi FB, Volta U. Anti-actin IgA antibodies in severe coeliac disease. Clin Exp Immunol. 2004 Aug;137(2):386-92. doi: 10.1111/j.1365-2249.2004.02541.x.

Reviewer 2 Report

Comments and Suggestions for Authors

Dr. Cerro’s group reported that PAM mice exposed to gliadin-rich diet exhibited functional changes like CD. Author assessed the impairment of immune system function and suggested that this mouse model could be used for in vivo CD model. Though results from this study such as impaired immune system function, an elevated oxidative state in these mice are very like what we have seen in CD patients, certain key immunological elements of CD disease have not been discussed or explored in this model, which impact the validity of using this model for studying CD disease.

11.     Missing control mice that are not PAM. The author’s hypothesis is based on the previous report that chronic physiological stress is related to the CD development.  Need to include the control mice and fed them on the same diet to validate in this model, the CD-like symptom is stress-induced.

22.  Line 124-125. Please clarify the composition of the control diet and gliadin-rich diet. Was the gliadin-rich diet modified on the base of the control diet? And how was the 120g/kg determined? Average Daily Intake? Is that number relative to human exposure?

33.   To apply this model for CD research, it is vital to test if the mice have developed the disease.  Need anti-tTG IgA ELISA test data to prove these mice are experiencing CD.

44. Line 136. Celiac disease mainly occurs in the gut system. Rationale of studying immune cells from peritoneal cavity is not clear. Splenocytes only provide the information of inflammation at the systemic level. Need to provide the info of  immune cells from the site specific -small intestine.

55. Author explored the NK activity and proliferative capacity of peritoneal lymphocytes. However, the key factors of the development of the disease were not explored in this study. Gliadin/gluten-specific t cells, Tregs, IEL T cells need to be explored in this study.

Author Response

Comments and Suggestions for Authors

Dr. Cerro’s group reported that PAM mice exposed to gliadin-rich diet exhibited functional changes like CD. Author assessed the impairment of immune system function and suggested that this mouse model could be used for in vivo CD model. Though results from this study such as impaired immune system function, an elevated oxidative state in these mice are very like what we have seen in CD patients, certain key immunological elements of CD disease have not been discussed or explored in this model, which impact the validity of using this model for studying CD disease.

  1. Missing control mice that are not PAM. The author’s hypothesis is based on the previous report that chronic physiological stress is related to the CD development. Need to include the control mice and fed them on the same diet to validate in this model, the CD-like symptom is stress-induced.

Thank you for your comment. We chose to focus on the PAM group because they exhibit an inadequate stress response and anxiety state, which is central to our hypothesis about anxiety and stress-related immune dysfunction and its potential link to gastrointestinal pathologies, like celiac disease. Unlike typical control mice with a healthy stress response, PAM provide a unique model to study how those conditions and immune dysregulation influence gliadin-induced impairments. We did not include the non-prematurely aging mice (NPAM) with normal stress responses, as they are extraordinarily healthy, and we believed they would not reflect a population at risk of being affected by gliadin ingestion. However, we appreciate the value of broader control groups and will consider this suggestion in future studies to further validate our findings.

  1. Line 124-125. Please clarify the composition of the control diet and gliadin-rich diet. Was the gliadin-rich diet modified on the base of the control diet? And how was the 120g/kg determined? Average Daily Intake? Is that number relative to human exposure?

Thank you for your questions.

  1. Composition of the diets: The control diet used in the study was the A04 diet from Panlab, which is a standard rodent diet. The gliadin-rich diet was formulated by modifying the control diet, adding 120 g/kg of gliadin (a gluten protein) to the base diet. The rest of the composition was identical to the control diet, ensuring that any observed effects were solely due to the inclusion of gliadin. The gliadin was sourced and incorporated into the diet as a powdered supplement to achieve the final concentration.
  2. Determination of the 120 g/kg dosage: The concentration of 120 g of gluten per kg of food in the diet was chosen based on previous studies investigating the effects of gluten in animal models. This dose was selected to produce a significant physiological impact on immune function and redox balance without being excessive. It is important to note that the normal amount of gluten typically added to a mouse's diet is around 5%, so with this dose, we aimed to represent a gluten-rich diet. The dose of 120 g of gluten per kg of food corresponds to a diet containing 12% gluten, which is similar to the amount found in human food. For instance, regular wheat flour contains between 10% and 12% gluten. This gluten concentration in the diet was used to assess its effects on immune function and oxidative stress, as it is representative of gluten levels in certain processed foods in the human diet [38-40].
  3. Average Daily Intake and Human Relevance: The 120 g/kg represents the amount of gliadin per kilogram of the diet, but it is not directly based on human exposure. To determine the average daily intake in mice, we considered their daily food consumption, which was approximately 5–6 grams per mouse. Based on this, each mouse would be consuming around 0.6–0.72 grams of gliadin per day (120 g/kg * 5–6 grams of food), which is a relatively high, but not extreme, dietary exposure for this model.

While the exact conversion to human exposure is complex due to differences in metabolism, body size, and diet composition, the gliadin dose used here is within a reasonable range to mimic a gluten-rich diet in rodents and potentially reflect aspects of gluten exposure seen in human populations, especially for those who have a higher consumption of gluten-containing foods (Journal of basic and clinical pharmacy, 7(2), 27–31. https://doi.org/10.4103/0976-0105.177703). Future studies could explore further standardization of gliadin intake to more closely align with human dietary habits.

We hope this clarifies the diet formulation and rationale behind the gliadin dose. To clarify these aspects, all this information has been included in the revised version of our work:

“2.3. Diet: Gliadin was provided by PhD. Yolanda Sanz. The gliadin-rich diet was prepared by supplementing the standard control diet with 120 g/kg of gliadin while maintaining the same overall nutritional composition. This approach ensured that any observed effects were specifically due to gliadin inclusion. The selected concentration was based on previous studies, aiming to induce significant changes in immune function and redox balance without reaching excessive levels. Standard rodent diets typically contain about 5% gluten, whereas this formulation provided a gluten-rich diet with 12% gluten, comparable to levels found in human foods such as wheat flour (10–12%). This dietary composition allowed for a relevant evaluation of gliadin’s impact on immune function and oxidative stress, reflecting gluten levels present in certain processed foods consumed by humans [38-40]”.

[38] Zhang L, Andersen D, Roager HM, Bahl MI, Hansen CH, Danneskiold-Samsøe NB, Kristiansen K, Radulescu ID, Sina C, Frandsen HL, Hansen AK, Brix S, Hellgren LI, Licht TR. Effects of Gliadin consumption on the Intestinal Microbiota and Metabolic Homeostasis in Mice Fed a High-fat Diet. Sci Rep. 2017 Mar 16;7:44613. doi: 10.1038/srep44613. PMID: 28300220; PMCID: PMC5353615.

[39] Olivares M, Rodriguez J, Pötgens SA, Neyrinck AM, Cani PD, Bindels LB, Delzenne NM. The Janus Face of Cereals: Wheat-Derived Prebiotics Counteract the Detrimental Effect of Gluten on Metabolic Homeostasis in Mice Fed a High-Fat/High-Sucrose Diet. Mol Nutr Food Res. 2019 Dec;63(24):e1900632. doi: 10.1002/mnfr.201900632. Epub 2019 Nov 11. PMID: 31608562; PMCID: PMC7003472.

[40] Aguilar EC, Navia-Pelaez JM, Fernandes-Braga W, Soares FLP, Dos Santos LC, Leonel AJ, Capettini LDSA, de Oliveira RP, de Faria AMC, Lemos VS, Alvarez-Leite JI. Gluten exacerbates atherosclerotic plaque formation in ApoE-/- mice with diet-induced obesity. Nutrition. 2020 Jul-Aug;75-76:110658. doi: 10.1016/j.nut.2019.110658.

  1. To apply this model for CD research, it is vital to test if the mice have developed the disease. Need anti-tTG IgA ELISA test data to prove these mice are experiencing CD.

Thank you for your comment. We agree that it is crucial to test if mice have developed celiac disease through the determination of localized inflammation and immune cell activation in the small intestine, intestinal villi deterioration, cytoarchitecture, and epithelial integrity, as well as immune cell infiltration analyses. Moreover, the assessment of specific CD markers should be also considered. For that reason and trying to avoid these possible confusions, we have revised the focus of the manuscript in the updated version. We have decided to be more cautious in our conclusions and no longer propose our model as a potential in vivo model for celiac disease. Indeed, this information has also been included as a limitation in the discussion section. In future research, we plan to investigate intestinal permeability and histological signs of mucosal damage to confirm the disease-like features and validate these PAM as a potential celiac mouse model.

  1. Line 136. Celiac disease mainly occurs in the gut system. Rationale of studying immune cells from peritoneal cavity is not clear. Splenocytes only provide the information of inflammation at the systemic level. Need to provide the info of immune cells from the site specific -small intestine.

Thank you for your comment. We understand the importance of studying immune cells at the site of the disease, particularly the small intestine, in the context of celiac disease (CD). While our current study focused on systemic immune responses using peritoneal, splenic, and thymic leukocytes, we recognize that assessing immune cells directly from the small intestine will be crucial for validating our model in relation to CD.

In response to your feedback, we have decided to be more cautious in our conclusions. We have revised the focus of the manuscript and no longer propose our model as a potential “in vivo” model for celiac disease. Future studies will need to explore key features of CD, such as localized inflammation and immune cell activation in the small intestine, to better validate our model in the context of CD. We appreciate your valuable input and will incorporate these considerations into our future work. Also, this information has been already included as a limitation of our study in the discussion section of our manuscript as he/she can see in the revised version:

“It is important to highlight some limitations of this study. First, although celiac disease (CD) has a higher incidence in women than in men [69-71], and for this reason, we used female PAM in this work, future research should also include male mice with premature aging. This would help determine whether these animals when fed a glia-din-rich diet, exhibit similar impairments to those observed in female PAM. Second, while our study focused on evaluating immune function parameters and oxidative stress markers, we recognize that key features of CD need to be further explored to better validate our model. Future experiments will incorporate laboratory tests to assess localized inflammation and immune cell activation in the small intestine, intestinal villi deterioration, cytoarchitecture, and epithelial integrity, as well as immune cell infiltration analysis. Moreover, the assessment of specific CD markers, such as an-ti-tissue transglutaminase, anti-endomysial, and anti-deamidated gliadin peptide antibodies [72,73] or serum anti-actin antibodies [74] will be also included in future investigations to further strengthen the translational relevance of our model.”

[69] Ivarsson A, Persson LA, Juto P, Peltonen M, Suhr O, Hernell O. High prevalence of un-diagnosed coeliac disease in adults: a Swedish population-based study. J Intern Med. 1999 Jan;245(1):63-8. doi: 10.1046/j.1365-2796.1999.00403.x.

[70] Gomez JC, Selvaggio GS, Viola M, Pizarro B, la Motta G, de Barrio S, Castelletto R, Echeverría R, Sugai E, Vazquez H, Mauriño E, Bai JC. Prevalence of celiac disease in Argentina: screening of an adult population in the La Plata area. Am J Gastroenterol. 2001 Sep;96(9):2700-4. doi: 10.1111/j.1572-0241.2001.04124.x.

[71] Rutz R, Ritzler E, Fierz W, Herzog D. Prevalence of asymptomatic celiac disease in ado-lescents of eastern Switzerland. Swiss Med Wkly. 2002 Jan 26;132(3-4):43-7. doi: 10.4414/smw.2002.09793.

[72] Singh A, Pramanik A, Acharya P, Makharia GK. Non-Invasive Biomarkers for Celiac Disease. J Clin Med. 2019 Jun 21;8(6):885. doi: 10.3390/jcm8060885.

[73] Ramírez-Sánchez AD, Tan IL, Gonera-de Jong BC, Visschedijk MC, Jonkers I, Withoff S. Mo-lecular Biomarkers for Celiac Disease: Past, Present and Future. Int J Mol Sci. 2020 Nov 12;21(22):8528. doi: 10.3390/ijms21228528.

[74] Granito A, Muratori P, Cassani F, Pappas G, Muratori L, Agostinelli D, Veronesi L, Bortolotti R, Petrolini N, Bianchi FB, Volta U. Anti-actin IgA antibodies in severe coeliac disease. Clin Exp Immunol. 2004 Aug;137(2):386-92. doi: 10.1111/j.1365-2249.2004.02541.x.

  1. Authors explored the NK activity and proliferative capacity of peritoneal lymphocytes. However, the key factors of the development of the disease were not explored in this study. Gliadin/gluten-specific t cells, Tregs, IEL T cells need to be explored in this study.

Thank you for your insightful comment. We acknowledge that key factors in the development of celiac disease (CD), such as gliadin/gluten-specific T cells, Tregs, and intraepithelial lymphocytes (IELs), play a crucial role in understanding the CD pathophysiology. In our case, this work supposes the first approximation of possible alterations that can promote gliadin in an “in vivo” experiment employing PAM. For this reason, we focused on evaluating immune function and redox alterations in immune cells from three different immune locations: the peritoneal cavity, spleen, and thymus, with the main goal to know whether this gliadin-rich diet could produce a general inflammation, and immune deterioration, both characteristics of celiac disease [7,9,10]. Also, all these parameters in which we have expertise allowed us to characterize PAM as a model for studying premature aging associated with inadequate stress response and anxiety [34-37].

Indeed, given that our original aim was to explore systemic immune and oxidative stress responses in prematurely aging mice (PAM) exposed to a gliadin-rich diet, we did not specifically examine intestinal immune responses, such as gluten-specific T cells and IELs, in this phase of our research.

After considering your valuable feedback, we have decided to be more cautious in our conclusions. Since our study did not address key aspects of CD, particularly localized immune responses in the small intestine, we no longer propose our model as a direct in vivo model for CD. Future studies will focus on investigating gluten-specific immune cells, Tregs, and IELs, which are fundamental to the development of the disease, to provide a more comprehensive understanding of CD pathophysiology.

[7] Marafini I, Monteleone I, Di Fusco D, Sedda S, Cupi ML, Fina D, Paoluzi AO, Pallone F, Monteleone G. Celiac Disease-Related Inflammation Is Marked by Reduction of Nkp44/Nkp46-Double Positive Natural Killer Cells. PLoS One. 2016 May 12;11(5):e0155103. doi: 10.1371/journal.pone.0155103.

[9] Freeman HJ. Pancreatic endocrine and exocrine changes in celiac disease. World J Gastroen-terol. 2007 Dec 21;13(47):6344-6. doi: 10.3748/wjg.v13.i47.6344.

[10] Pop AV, Popa SL, Dumitrascu DL. Extra-digestive manifestations of celiac disease. Med Pharm Rep. 2024 Jul;97(3):249-254. doi: 10.15386/mpr-2776.

[34] Pérez-Alvarez L, Baeza I, Arranz L, Marco EM, Borcel E, Guaza C, Viveros MP, De la Fuente M. Behavioral, endocrine and immunological characteristics of a murine model of premature aging. Dev Comp Immunol. 2005;29(11):965-76. doi: 10.1016/j.dci.2005.02.008.

[35] Viveros MP, Arranz L, Hernanz A, Miquel J, De la Fuente M. A model of premature aging in mice based on altered stress-related behavioral response and immunosenescence. Neuroimmunomodulation. 2007;14(3-4):157-62. doi: 10.1159/000110640.

[36] Garrido A, Cruces J, Ceprián N, Vara E, de la Fuente M. Oxidative-Inflammatory Stress in Immune Cells from Adult Mice with Premature Aging. Int J Mol Sci. 2019;20(3):769. doi: 10.3390/ijms20030769.

[37] Félix J, Díaz-Del Cerro E, Garrido A, De La Fuente M. Characterization of a natural model of adult mice with different rate of aging. Mech Ageing Dev. 20024;222:111991. doi: 10.1016/j.mad.2024.111991.

Round 2

Reviewer 2 Report

Comments and Suggestions for Authors

I agree with the current version.